

# Local and regional flood frequency analysis based on hierarchical Bayesian model: application to annual maximum streamflow for the Huaihe River basin

Yenan Wu[1], Upmanu Lall[2, 3], Carlos H.R. Lima[4], Ping-an Zhong[1]

[1]College of Hydrology and Water Resources, Hohai University, Nanjing, 210098, China

[2]Columbia Water Center, Columbia University, New York, NY, 10027, USA;

[3]Department of Earth and Environmental Engineering, Columbia University, New York, 10027, USA

[4]Civil and Environmental Engineering, University of Brasilia, Brasilia, 70910-900, Brazil

*Correspondence to: Ping-An Zhong(pazhong@hhu.edu.cn)*

**Abstract.** We develop a hierarchical, multilevel Bayesian model for reducing uncertainties in local (at-site) and regional (ungauged or short data sites) flood frequency analysis. This model is applied to the annual maximum streamflow of 17 gauged sites in the Huaihe River basin, China. A Generalized Extreme Value (GEV) distribution is considered for each site, and its location and scale parameters depend on the site's drainage area. We assume the hyper-parameters come from Non-informative (independent, uniform) prior distribution and sample values from posterior distribution by the MCMC method using Gibbs sampling. For comparison, the ordinary GEV fitting by Maximum Likelihood Estimate (MLE) and index flood method fitted by L-moments are also applied. The local simulation results show that for most sites the 95% credible interval simulated by the Hierarchical Bayesian model are narrower than the at site GEV outputs thus reducing uncertainty. By comparison, the homogeneity assumption of the index flood method often leads to large deviations from the empirical flood frequency curve. Cross validated flood quantiles and associated uncertainty intervals are also derived. These results show that the proposed model can better estimate the flood quantiles and their uncertainty than the index flood method.

## 1 Introduction

Floods are one of most destructive climate-related disasters (Hirabayashi et al., 2013). China (Wu et al., 2015; Yin et al., 2016), central Europe (Schröter et al., 2015), the United States (Mallakpour and Villarini, 2015), United Kingdom (Van Oldenborgh



et al., 2015), and many other countries (Filizola et al., 2014) incurred flood disasters in recent years which led to humanitarian

crises and economic losses (Steinschneider and Lall, 2015). As a highly developed region of agriculture and industry, Huaihe

River basin has become one of most important food and energy bases in China (Yang et al., 2012). The basin is located in the

climate transition zone of subtropical and warm temperate zones. About 30% of the Huaihe River basin area, the northeast,

west and southwest of basin boundaries is upland and headwaters. The other 70% is low plains with numerous lakes. Due to

its climate and geography, the Huaihe River basin frequently experiences flood disasters (Jin et al., 2015). Consequently, 39

large reservoirs and flood detention basin have been designed to improve the flood control capacity. However, flood disasters

are still severe in the Huaihe River basin. Three basin-wide floods happened in the Huaihe River basin at the year of 1991,

2003, and 2007. These three floods were the most destructive flood events in the historical record and caused 609 death and

281 million of collapsed building as well as an economic losses of 744 billion yuan (Zhang and You, 2014). Since, the observed

streamflow data series across all gauged sites in China are generally short. It is difficult to obtain good estimates of the design

flood using data series for a single site. Therefore, reduced uncertainties of flood frequency estimates for the Huaihe River

basin is of great importance for the implementation of flood control and reservoir operation.

        Regional frequency analysis (RFA) has been used to reduce uncertainties for poorly gauged sites or ungauged sites

(Halbert et al., 2016) by using data from multiple sites, many of which may have better, longer records. RFA enables reduction

the uncertainties in estimating frequency at ungauged site or sites with short records (Sun et al., 2015). Regional models differ

in the way used to transfer information (Ganora and Laio, 2016) across the region. Wagener (Wagener et al., 2013) notes that

there are two main ways to transfer information. One way is to cluster sub-basins based on a variety of sub-basin attributes

(e.g. elevation, slope, drainage area or land cover type), since similar sub-basins are expected to have a similar hydrological

condition. Each such cluster is considered to be homogeneous region and has a similar flood formation mechanism. Thus, the

variable of interest (e.g. maximum streamflow) can be drawn from a same probability distribution across the region. Predictions

at an ungauged site are performed using the hydrological information at similar gauged sites to estimate the variable of interest

at ungauged site. A second way is to develop a relationship between the hydrological variable of interest and sub-basin

attributes, which is called a mapping function (expressed as a regression model). Prediction is achieved by applying the

regression relationship to estimate the parameters for the ungauged site, with the known attribute covariates from the gauged

sites.

Many regional flood frequency analysis models have incorporated temporal or/and spatial covariates for the mapping function (Chen et al., 2014; Lima and Lall, 2010; Renard and Lall, 2014; Sun et al., 2015). The temporal covariates are often identified as time or climatic indices (e.g. El-Nino-Southern Oscillation (ENSO) indices), while site characteristics (e.g. drainage area, elevation) are considered as covariates in space. Of a variety of choices, the index flood method has been widely applied because of its easy implementation and robustness (Renard, 2011). The index flood approach uses the mean or median of at-site hydrological variable data (Kjeldsen et al., 2002; Kjeldsen and Jones, 2006), and uses it to normalize the data series of each site. The resulting series are using a regression model linking the resulting flood series and site attributes. The main assumptions of this approach are 1) the scale invariance, i.e., considering the whole region as a homogenous one; and 2) that the regression function (e.g. ordinary least squares) may provide the estimation error for the index flood value. Stedinger and Tasker (Stedinger and Tasker, 1986, 1985) proposed an improved method using generalized least squares to consider the estimation errors. Kjeldsen and Jones (Kjeldsen and Jones, 2009) considered the regression error on spatial correlation in generalized least squares approach. These approaches still assume scale invariance.

A Hierarchical Bayesian (HB) model has been investigated more recently (Kwon et al., 2008; Yuan et al., 2016) primarily because it relaxes the assumption of scale invariance and may better quantify both parameter and model uncertainties. Through a HB model, at-site information and regional dependence are considered together, and flood quantiles for different return periods can be estimated simultaneously. In this paper, we provide an application of a hierarchical, multilevel Bayesian model to analyze regional flood frequency and reduce uncertainties at both estimation of parameter and flood frequency prediction at ungauged sites or poorly gauged sites.

The model developed is applied using annual maximum streamflow series at 17 gauged sites in the Huaihe River basin, China. For comparison, flood frequency analysis based on ordinary GEV which parameters are estimated by maximum likelihood method and based on index flood method fitted by L-moments method are also applied. We clarify the assumptions made in both approaches and demonstrate the comparative advantage of the Hierarchical Bayesian approach in reducing at site and regional estimation uncertainties, while providing a more general and flexible modeling approach.

In the following parts, Section 2 provides the structure of the hierarchical Bayesian model and an example with specific





distribution and covariate for flood frequency analysis. An application in Huaihe River basin of China is presents in Section 3.

Ended with a summary in Section 4.

## 2 Hierarchical Bayesian model for flood frequency analysis

### 2.1 General modeling framework

A multilevel, Hierarchical Bayesian model that considers pooling of information across all gauged sites through a regression on covariates is considered. This pooling reduces the equivalent number of independent parameters, resulting in lower uncertainty in parameter estimates and therefore leads to reduced uncertainty in the flood frequency analysis. The approach is as follows:

Consider that $Y(i,j)$ represent the observed variable of interest (e.g. annual maximum flow) at site $i$ and time $j$.

$Y(i,j)$ is considered to follow a distribution whose parameters vary with space.

$$Y(i,j) \sim D\big(\boldsymbol{\varphi}(i)\big) \tag{1}$$

where $D$ is a specific distribution (e.g. Gumbel, GEV) and $\boldsymbol{\varphi}(i) = \big(\varphi^k(i)\big)_{k=1,\dots,K}$ is a vector of the parameters involved, where $K$ is the number of parameters. Note that the parameter vector $\boldsymbol{\varphi}(i)$ can be allowed to vary in time. The applications in this paper consider only spatial variation.

The multilevel can typically be set up to consider 3 settings: non-pooled, fully-pooled, and partially pooled (Devineni et al., 2013). In a non-pooled model, local parameters of each site are estimated separately. In a fully-pooled model, common parameters for all sites are considered. In a partially-pooled model, parameters of each individual site may different, but are assumed to be drawn from a common distribution. The cross site dispersion of the parameters is modeled explicitly using the regional data. Consider a regression model for the parameters across sites that is defined as follows:

$$\varphi^k(i) = R^k\left(\big(\psi_l^k\big)_{l=1,\dots,L}, X(i)\right) \tag{2}$$

where $R^k(\ )$ is a general regression function which can be linear or nonlinear; $X(i)$ denotes a set of covariates for each site.





The term $\psi_l^k$ is a vector of hyper-parameters associated with covariates $X(i)$ and $L$ is the number of hyper-parameters.

The $\psi_l^k$ are drawn from a distribution of hyper-parameters which is described as the second level of HB model:

$$\psi_l^k \sim D_l^{'}\left(\vec{\psi}_l\right) \tag{3}$$

where $\vec{\psi}_l$ is a vector of hyper-parameters, and $D_l'(\ )$ is the corresponding density function.

The likelihood function for each site $i$ of the model can be described by using Bayes theorem as follows:

$$f_H(i) = \prod_{j=1}^{J} f_D\left(y(i,j)\big|\varphi(i)\right)\prod_{l=1}^{L} f_{D_l^{'}}\left(\psi_l^k\big|\vec{\psi}_l\right) \tag{4}$$

The non-pooled model and the fully-pooled model are two special cases of the partially-pooled model. A structure of proposed model is shown in Fig.1.

### 2.2 Formulation

Based on the above framework, local and regional flood frequency analysis are explored using three models, M0, M1, and M2 that account for increasing information provided by site-specific or/and regional characteristics. The distribution of annual maximum flow at each site can be described as a general extreme value (GEV) distribution based on the goodness-of-fit tests (Table 1), which is widely applied in flood frequency analysis (Smith et al., 2015; Zhang et al., 2014). The three nested models are described as follows: the non-pooled model, M0, presents a local flood frequency analysis model, where the parameters

from each site are estimated separately by using traditional GEV distribution. Let $Y(i,j)$ represent annual maximum flow series for site $i$ in year $j$, which follow a GEV distribution given the location $\mu_i$, scale $\sigma_i$, and shape $\xi_i$ parameters at site $i$:

$$Y(i,j) \sim GEV\left(\mu_i, \sigma_i, \xi_i\right) \tag{5}$$

A maximum likelihood estimate (MLE) method is chosen for parameter estimation. Other methods (e.g. L-moments) are

described by (Kirshen et al., 2008; Noto and La Loggia, 2009).

The fully-pooled model, M1, is similar to a typical regional flood frequency model. The procedure entails four steps: (1)



calculate the index flood of all sites by taking the average or the median of the annual maximum flow series $IF_i$ and then

obtain the standardized annual maximum flow matrix $z(i,j)$ using the index flood of each site.

$$z(i,j) = \frac{Y(i,j)}{IF_i} \tag{6}$$

(2) L-moments are used to estimate the parameter of the common regional distribution and obtain the regional growth

curve $\omega_T$ ($T$ is a $T$-year event), based on the set of standardized at-site data.

(3) Build up the regression model linking the index flood $IF_i$ and site attributes $X(i)$ from gauged site for transferring

the index flood to ungauged site which has the known site attribute.

$$IF_i = R(X(i)) \tag{7}$$

(4) Finally, the flood frequency curve $F_i$ for site $i$ can be calculated by taking product of the regional growth curve.

$$F_i = IF_i \cdot \omega_T \tag{8}$$

The partially-pooled model, M2, allows the GEV parameters to vary by site based on the drainage area (Lima et al., 2016)

(other basin attributes like elevation, slope or land use could also be used). We consider a mapping function, a log-log linear

relationship, linking the prior distribution of GEV location $\mu_i$ and scale $\sigma_i$ parameter to the drainage area, which has been

demonstrated theoretically and empirically (Gupta and Dawdy, 1995; Morrison and Smith, 2002; Northrop, 2004; Villarini

and Smith, 2010).

$$p(\log(\mu_i)) \sim N(\alpha_0 + \alpha_1 \cdot \log(A_i), \tau_\mu^2) \tag{9}$$

$$p(\log(\sigma_i)) \sim N(\beta_0 + \beta_1 \cdot \log(A_i), \tau_\sigma^2) \tag{10}$$

where $A_i$ is the drainage area of site $i$, the terms $\alpha_0, \alpha_1, \beta_0, \beta_1, \tau_\mu^2, \tau_\sigma^2$ are hyper-parameters.

The shape parameter, $\xi_i$, has not previously been found to have a dependence on drainage area. In the previous studies,

the regional shape parameter usually obtained by averaging the local shape parameter at each site or using all the data to

estimate the parameter, since the shape parameter is often more difficult to estimate reliably. In this paper, we assume that the





shape parameter for each site is different but comes from a common Normal distribution with a mean and variance:

$$\xi_i \sim N\left(\overline{\xi}, \tau_\sigma^2\right) \tag{11}$$

where the parameters $\overline{\xi}$ and $\tau_\sigma^2$ of normal distribution are hyper-parameters. This is equivalent to considering a mean shape

parameter $\overline{\xi}$, but does allow some dispersion across sites that is informed by the actual data.

The hyper-parameters in equation (9)-(11) are assumed to come from a non-informative (independent, uniform) prior

distribution as suggested by Gelman (Gelman et al., 2014).

$$p\left(\alpha_0, \alpha_1, \beta_0, \beta_1, \overline{\xi}, \tau_\mu, \tau_\sigma, \tau_\xi\right) \propto 1 \tag{12}$$

Bayes' rule is used for writing an expression of the posterior probability distribution for all the parameters:

$$p\left(\Lambda|q\right) = \frac{p\left(q|\Lambda\right) \cdot p\left(\Lambda\right)}{p\left(q\right)} \propto p\left(q|\Lambda\right) \cdot p\left(\Lambda\right) \tag{13}$$

where $q$ is the observed annual maximum flow data; $\Lambda = \left[\mu, \beta, \sigma, \alpha_0, \alpha_1, \beta_0, \beta_1, \overline{\xi}, \tau_\mu, \tau_\sigma, \tau_\xi\right]$ is the whole parameters

in this model; $p\left(\Lambda\right)$ refers to the prior distribution of all the parameters; the denominator $p\left(q\right)$ is a normalized factor and

$p\left(q|\Lambda\right)$ is the likelihood function:

$$p\left(q|\Lambda\right) = \prod_{i=1}^{I} \prod_{j=1}^{J_i} GEV\left(y_{ij}|\mu_i, \sigma_i, \xi_i\right) \tag{14}$$

where $y_{ij}$ is the observed annual maximum flow data for site $i$ in the year $j$; $J_i$ is the number of years has observed data

in site $i$; $I$ is the number of sites in the study area of interest. Note that each site may have different length of available data.

     Substituting the likelihood function Eq. (14) and prior distributions of Eq.(9)-(12) into the Bayes' equation, then we get

the posterior distribution of the parameter vector $\Lambda$ :

$$p\left(\Lambda|q\right) \propto \prod_{i=1}^{I} \prod_{j=1}^{J} GEV\left(y_{ij}|\mu_i, \sigma_i, \xi_i\right) \cdot N\left(\log\left(\mu_i\right)|\alpha_0 + \alpha_1 \cdot \log\left(A_i\right), \tau_\mu^2\right) \cdot$$
$$N\left(\log\left(\sigma_i\right)|\beta_0 + \beta_1 \cdot \log\left(A_i\right), \tau_\sigma^2\right) \cdot N\left(\xi_i|\overline{\xi}, \tau_\sigma^2\right) \tag{15}$$

A major challenge in the Bayesian model is to calculate integrals for the posterior distribution of all the parameters in the

model (Khan and Coulibaly, 2010). The dimensionality of the integrals is determined by the number of parameters, and

modeling the joint posterior distribution across all the parameters is usually not tractable. In this paper we use Markov Chain

Monte Carlo (MCMC) sampling, which is widely used in Bayesian method to draw values of the set of parameters (Gaume et

al., 2010; Reis and Stedinger, 2005). In particular, Gibbs sampler is used to sample values of both hyper-parameters and

parameters of GEV distribution from Eq. (15) (Lima and Lall, 2010). Three chains for MCMC simulation are run to verify the

convergence of the results and 1500 simulations of the joint posterior distribution of the hyper-parameters are drawn to verify

the reproduction ability of the model to estimate the flood frequency in ungauged sites.

In the HB process, simulations of the local flood quantile for different return periods and their associated uncertainty

intervals can be obtained by drawing samples of the location $\mu_i$ ,scale $\sigma_i$ and shape $\xi_i$ parameters from the posterior

distribution of parameter vector $\Lambda$ using Eq.(5). These are compared with the results of ordinary GEV across each site fitting

by maximum likelihood method (M0) and with the results of the index flood method fitting by L-moments (M1). Prediction

of flood quantiles and associated uncertainty intervals for ungauged site can be obtained by drawing samples of the hyper-

parameters from the joint posterior distribution and then using the Eq. (9)-(11). The flood quantile of different return periods

(regional flood frequency) for the predicted site can be obtained by using Eq.(5) with the sampled GEV parameters. For

comparison, the regional flood frequency analysis via index flood method fitting by L-moments method also included for each

ungauged site (M1).

## 3 Case study

### 3.1 Study area and data

The proposed model is applied to a flood-prone area of China, the Huaihe River basin, which is located between Yellow River

basin and Yangtze River basin (see Fig.2)(Wu et al., 2016). Annual maximum streamflow data with a total of 17 gauge sites

distributed in both sides of the Huaihe River are provided by the Huaihe River Commission of the Ministry of Water Resources.

The data are gathered over the period from 1950 to 2015, but none of the sites have a complete record. Site #15 has the longest





record with 64 years while site #17 has the shortest one with 27 years of record. The drainage area ranges from 185 to 133950

km² (see Table 2 for details).

### 3.2 Results

Sites #12 and #16 were randomly chosen as the cross-validation sites and are not included in the HB estimation procedure.

The fitted model is then used to predict the flood frequency for these two sites. In this way, we can test the efficiency of the

proposed model in using data to predict the flood frequency in ungauged sites or with missing value. We first explore whether

the at site estimates of the GEV parameters relate to drainage area as assumed in methodology. We draw a scatter plot between

the logarithm of drainage area for each site and the corresponding logarithms of GEV estimates of location, scale, and shape

parameters estimated by MLE and then fit an ordinary least squares (OLS) regression function for each parameter as shown in

Fig.3. We note that a log-log relationship between the location and scale parameters of annual maximum streamflow series

and drainage area is supported. The coefficient of determination ($R^2$) of the regression function associated with location and

scale parameter are 0.88 and 0.80, respectively. As expected, the shape parameter does not support a scaling relationship with

the drainage area. In this the slope from the OLS regression is negative and the $R^2$ value is relatively low (0.0226). Finally, a

nonparametric test, K-S test (Chowdhury et al., 1991), is used to check the hypothesis that the estimated shape parameter is

normally distributed. This hypothesis is not rejected at a significance level $\alpha = 0.05$.

The boxplot in Fig.4 shows the distribution of GEV parameters simulated by the HB model compared with the original

GEV parameters estimated by MLE (red dots) across all sites (cross-validation sites are excluded). The distribution of GEV

parameters estimated by HB model contains all the MLE estimations while most of MLE estimations of location and scale

parameters are located in the "box" (between first quantile and third quantile) of HB estimations. Shape parameters for most

of simulation sites show a large variance reflected as long whiskers and the results suggested that the annual maximum

streamflow data series for most sites may have a heavy tail with shape parameters larger than 0. More than half of shape

parameters estimated by MLE are not in the "box" of parameter distribution by HB model, since it is hard to obtain a precise

estimation with short data records. It is interesting to note that the site #11 which has only 31 years of data has a relatively



large uncertainty on the distribution of location, scale and shape parameters while for site #15 whom has 64 years records shows a low uncertainty.

Figure 5 shows the estimation of flood quantile with different return periods for each of the models M0, M1, and M2 across six randomly selected sites (site #4, #5, #8, #11, #14, #15). The black line is the result of flood quantile estimated by ordinary GEV (M0) and the dashed black line is the associated 95% credible interval. The red line and the pink region are the results of median of posterior distribution of HB model and associated 95% credible interval (M2), respectively. The blue line is the flood frequency estimated by the index flood method (M1). For comparison the empirical estimates of flood frequency are shown as black dots in Fig.5. For all the selected sites, the lower boundary of the credible interval estimated by M0 tends to reduce as the return periods increase, which shows the difficulty in using this method to estimate the credible interval especially for the extreme events with large return periods. In partially-pooled model, M2, the credible interval (pink region) contains most of empirical estimates and is generally narrower than the results of ordinary GEV, highlighting the uncertainty reduction of the HB method. In the fully-pooled model, M1, the information across sites are combined together, but the region may not be homogeneous. Hence the estimation with this assumption may lead to large bias, seen from the results of sites #4, #8, and #14, the results of index flood method (blue line) are large deviated from the ordinary GEV and HB results, which shows the large uncertainty when using the index flood method to predict the flood frequency for ungauged site. Overall, the result of hierarchical Bayesian model (M2) show its advantage for flood frequency estimation and uncertainty reduction especially for the extreme events with large return periods that out of record.

The streamflow data of site #12 and #16 are selected as cross-validation samples for assessing the performance of proposed HB model for predicting the flood frequency at ungauged sites (regional flood frequency analysis), which is not used in the Bayesian simulation procedures. We use just the drainage area of the two sites and then the parameter distribution of GEV are obtained from the posterior distribution of hyper-parameters associated with the mapping relationship of drainage area. The prediction results are shown in Fig.6. The flood quantile estimated by the median of posterior distribution of HB model and their associated 95% credible interval are presented as red line and pink shade area. For comparison purpose, empirical estimates of the measured streamflow data and index flood method are also developed and illustrated by black dots and blue lines, respectively. For sites #12 and #16, the HB credible interval contains all the empirical estimates and flood

quantile estimates by predicted median of posterior distribution of HB model (red line) fits the empirical ones well especially in site #16. Although the predicted median of posterior distribution of HB model has some deviation from the empirical estimates (black dots) in site #12, it still significantly improved from the results of index flood method fitted by the L-moments.

## 4. Summary

In this paper, we demonstrated a hierarchical, multilevel Bayesian model for improving at-site (local) and regional (ungauged site or lack of record) flood frequency analysis and reducing parameter uncertainty. The application across 17 gauged sites (15 sites for Bayesian estimation procedure and 2 cross-validated sites for prediction) in the Huaihe River basin show the substantial advance in RFA and properly pooling information.

The mapping function between two GEV parameters (location and scale parameters) with drainage area provides effective information transfer in space. The distribution assumption of shape parameter in this case appears to be an effective way to let the shape parameter can be able to different by site but follow a certain law. This can improve the drawbacks that the non-pooled model may lead to large bias in sites with small record and the fully-pooled model which the homogeneity assumption may be too restrictive in some cases.

In the case study, three nested models, non-pooled (M0), fully-pooled (M1), and partially-pooled (M2) were compared and the partially-pooled model was demonstrated to have the best performance. The fully-pooled model following the index flood method with L-moments has large uncertainty in simulation and prediction due to the assumption of homogeneity. The result of the proposed Hierarchical Bayesian model is generally better and the substantial gain of this partially-pooled model is in estimating flood quantiles of different return periods and associated credible interval. This model also provides a better way for using fragmented data with varying lengths for analyzing spatial distribution of flood frequency across multi-sites, and in that context provides an alternative to Stedinger's Generalized Least Squares approach.

In future work we plan to explore how to deal with small upland/headwater drainages and the larger drainage areas mechanistically. The flood frequency curve in different sizes of drainage areas in Huaihe River basin may not be the same since the larger basin may respond more to the larger scale dynamics of the monsoon or to the presence of typhoons, while for smaller one flash flooding may also dominated by local convection events.



The Hierarchical Bayesian model proposed here is suitable for other probability distribution and is flexible enough to consider climate variables or basin attributes other than the drainage area. An extension would be to add appropriately selected climate variables (eg. ENSO) and basin attributes (eg. elevation) simultaneously for analyzing the spatial-temporal distribution of flood frequency regionally and locally to address climate change and natural climate variability as well as other factors lead to nonstationary on probability distribution of hydrological variables. Our future work will focus on seasonal-ahead or yearly-ahead forecasts by using climate variables (e.g. NAO, AMO show significant correlation with streamflow in Huaihe River basin (Chen et al., 2014)) as covariates to make dynamic and adaptive rules for operating reservoir or other flood control project systems and keep flood risk remain unchanged. In this context simultaneous consideration of the joint distribution of flood volume maxima as well as instantaneous peak flow maxima using copulas may be appropriate (Grimaldi and Serinaldi, 2006).

**Acknowledgements.** This study was supported by the National Key R&D Program of China (Grant No. 2017YFC0405606), the National Natural Science Foundation of China (Grant No. 51579068), Postgraduate Research & Practice Innovation Program of Jiangsu Province (Grant No. KYCX17_0421), the Fundamental Research Funds for the Central Universities (Grant No.2017B616X14). The first author acknowledges financial support from China Scholarship Council.

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







**Figure 1: Schematic diagram of the proposed hierarchical Bayesian model.**





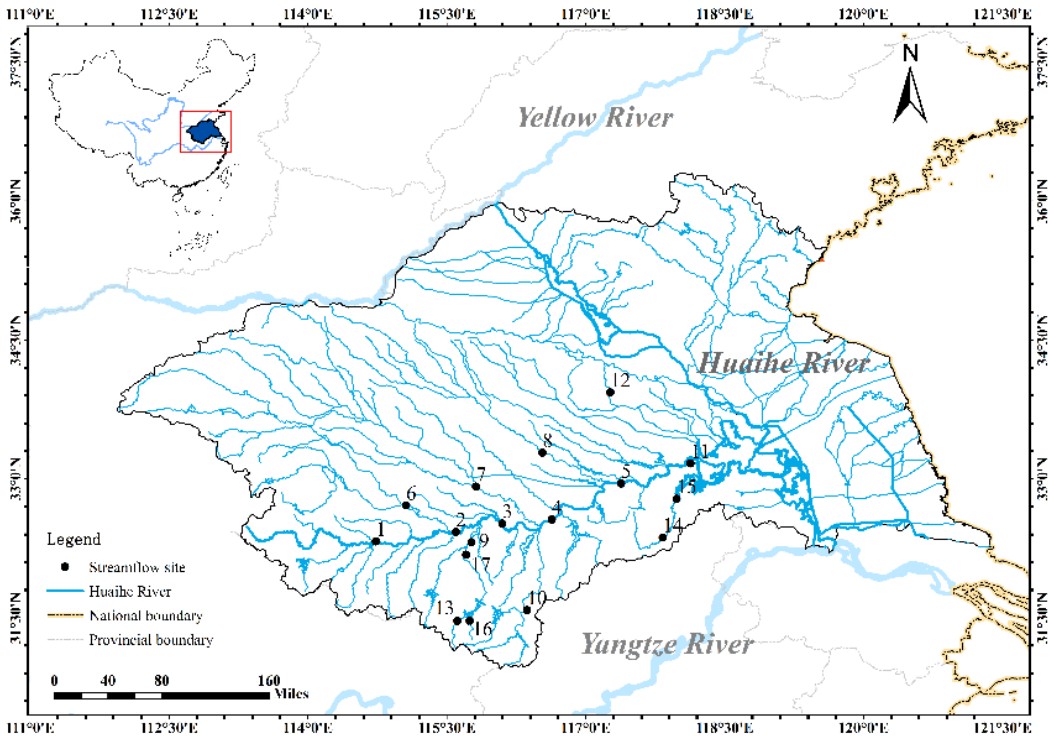

**Figure 2: location of Huaihe River basin in China and the distribution of streamflow gauged sites.**




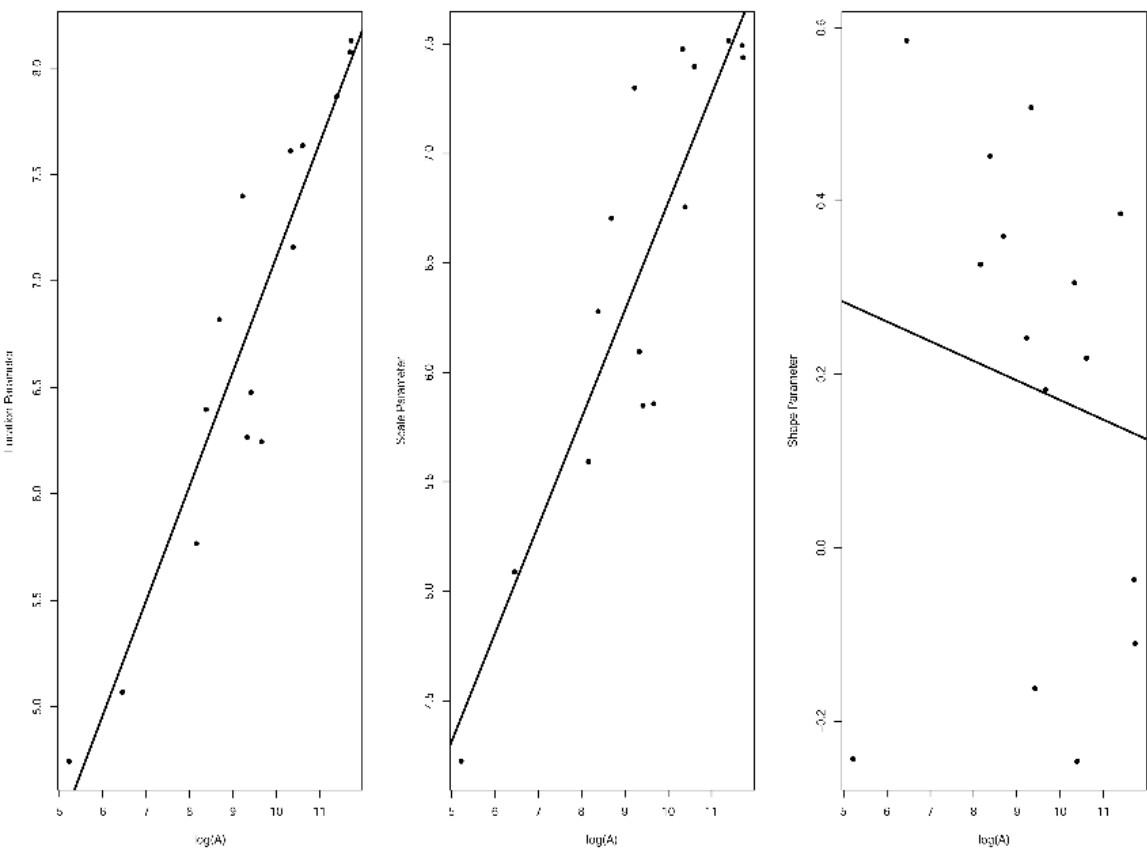

**Figure 3: The relation between the logarithm of drainage area across all sites (cross-validation sites are excluded) and the corresponding logarithm of GEV parameters. The black line represents the OLS regression.**





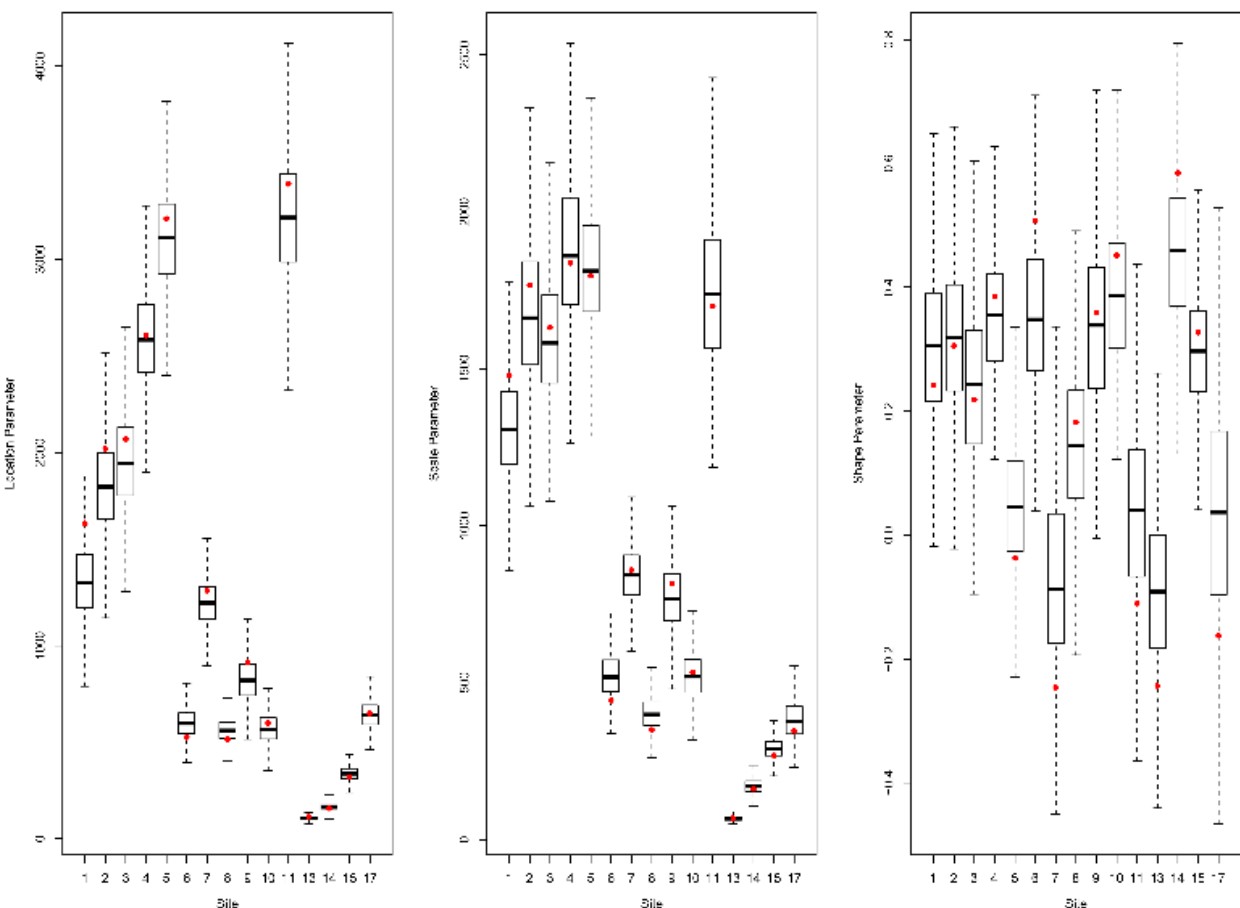

**Figure 4: GEV parameter distributions simulated by HB model compared with ordinary GEV parameter estimations by MLE (red dots) across all simulation sites.**





**Figure 5: Flood quantile estimation for selected sites based on M0, M1, and M2. The red line shows the median of posterior distribution of HB model (M2). The pink region represents the 95% credible interval for HB model. The black line and dash black line represent the flood frequency curve of ordinary GEV and associated 95% credible interval. The result of index flood method are the blue line. Black dots show the empirical frequency of annual maximum streamflow series.**





**Figure 6: Predicting flood quantile for the cross-validation sites #12 and #16 using index flood method (blue line) and HB model (red line). The pink region is the 95% credible interval of HB results while black dots denote the empirical estimation.**





**Table 1 goodness-of-fit test of annual maximum flow distribution based on the indices of negative log-likelihood, AIC, and BIC**

| Distribution | Negative Log-Likelihood | AIC | BIC |
|---|---|---|---|
| **GEV** | **375.08** | **756.74** | **761.65** |
| PEARSON III | 375.26 | 757.09 | 762.01 |
| GAMMA | 377.21 | 758.71 | 762.08 |
| NORMAL | 377.41 | 758.81 | 762.47 |
| POISSON | 377.41 | 758.81 | 762.47 |
| LOGNORMAL | 380.60 | 765.20 | 768.86 |
| EXPONENTIAL | 382.30 | 766.61 | 768.44 |

**Table 2 Drainage area and series length of each site used in this paper**

| Site | Drainage area (km²) | Series length (years) | Site | Drainage area (km²) | Series length (years) |
|---|---|---|---|---|---|
| 1 | 10143 | 55 | 10 | 4370 | 43 |
| 2 | 30630 | 54 | 11 | 123950 | 31 |
| 3 | 40360 | 55 | 12 | 3050 | 57 |
| 4 | 88630 | 56 | 13 | 185 | 49 |
| 5 | 121330 | 55 | 14 | 635 | 55 |
| 6 | 11280 | 53 | 15 | 3501 | 64 |
| 7 | 32486 | 56 | 16 | 1830 | 54 |
| 8 | 15745 | 52 | 17 | 12300 | 27 |
| 9 | 5930 | 55 | | | |

