# Peer review of "Local and regional flood frequency analysis based on hierarchical Bayesian model: application to annual maximum streamflow for the Huaihe River basin"

_Hydrology and Earth System Sciences, 2018_

## Referee Comment (RC1) · Anonymous Referee #1 · 27 Feb 2018

Summary: This study develops a hierarchical Bayesian model for regional flood frequency analysis in a region consisting of 17 annual maximum series in China. I would not consider the theoretical framework particularly novel as it is building heavily on previous work by the authors (e.g. Lima and Lall, 2010) without notable new innovations. At the same time, the comparisons with other methods is lacking somewhat in depth and rigor. In particular, I would recommend dropping the comparison with M1 as this model seems very inappropriate for the considered region. The paper does contribute to the knowledge of the flood hydrology of the considered region, but due to the before mentioned issues I do not think the manuscript contains sufficient originality and significance to be considered for publication in HESS.

[Figure]

Comments:

Page 2, line 14: The most common reason for using regional frequency analysis is to obtained estimates of design events at ungauged sites.

Page 2, line 15. Sentence starting "RFA enables..." should be deleted as it just repeats the statement made in the previous sentence.

Page 2, lines 17-25: I think the division of regionalization methods into homogeneous regions and regression link functions unnecessarily complicated. The regression model postulates a link between, say scale-parameter and drainage area, for example $\log(\sigma_i) = \beta_0 + \beta_1 \log(A_i)$. Then a model assuming a region to be homogeneous with regards to the scale parameter would simply assume $\beta_1 = 0$, so that $\log(\sigma_i) = \beta_0$. Therefore, the first approach (homogeneous region) is simply a variation on the more general regression approach.

Page 3, line 3: Three citations to the same group for something as common as using covariates in regional flood frequency analysis? This is excessive self-citation and much more representative references should be selected here (for example some of the classical USGS reports by Benson and Thomas on this topic).

Page 3, line 5-13: The discussion of scale invariance in the existing methods is a little confusing. Stedinger and Tasker developed their GLS framework by regressing design quantiles against covariates, so presumably scale-factors are implicitly considered. Kjeldsen and Jones (2009) developed their regression framework by focusing on the median annual maximum flood, so they did not consider scale at all.

Page 3, line 21: How does an 'ordinary GEV' differ from a 'GEV' distribution? Is there an 'abnormal GEV'? I suggest using 'a GEV distribution fitted directly to the at-site data' or something like that. The phrase 'ordinary GEV' is used through-out the manuscript.

Section 2.1: I found the model description quite abstract and difficult to follow. In fact, the description of the model in section 2.2 is much more straight forward and easy to

understand. Maybe try to incorporate the important parts from section 2.1 into a slightly more expanded version of section 2.2.

Page 4, line 12: I am not sure I understand the notation. $\varphi(i)=(\varphi\hat{\ }k\,(i))\_(k=1\ldots K)=(\varphi\hat{\ }1\,(i),\ldots,\varphi\hat{\ }k\,(i))\hat{\ }T$?

Eq. (3): In other parts of the manuscript, vectors are represented by bold characters, but here the "->" notation is adopted. I suggest being consistent and use bold notation.

Eq. (3): what is the significance of the apostrophe in D' ?

Eq. (4): This equation appears a little sudden, so maybe a bit more explanation might be useful. What does the subscript H signify? Is it not more common to denote a likelihood function by L rather than f?

I am not sure I really understand Figure 1. Is K always equal to L?

Table 1: As you have 17 data series, what does the AIC scores refer to?

Table 1: Very unusual to see the normal distribution performing better than the log-normal distribution. Please check this is correct?

Table 1: How can the Poisson distribution be considered a candidate distribution? It is a discrete distribution, but the annual maximum series are continuous random variables? This seems inappropriate to me.

Page 5, line 15-16: Y(I,j) has already been defined on page 4.

Page 5, line 21: by "a typical regional frequency model" I assume you mean the index flood method as presented by Hosking and Wallis (1997)?

Eq. (12): Not sure I understand this notation. Is this a way of writing a uniform distribution?

Page 7, line 10: Observed annual maximum flow is denoted q here, but Y in other parts of the manuscript?

[Figure]

Page 10, line 10: Should $\mu, \beta, \sigma$ be part of the $\Lambda$ vector?

Eq. (14): I can quite comprehend if the intersite dependence between annual maximum series across sites comes into this model formulation of if this formulation assumes data from the I sites to be independent. Note that the studies by Tasker and Stedinger and Kjeldsen and Jones discussed in the introduction consider the aspects of cross-correlation in data and model errors import parts of the development of regional methods. However, these aspects are not discussed here.

Page 7, line 14: Again, a change in notation from Y(i,j) to q and then on to yij. Personally I prefer the latter version.

Page 8, line 4: I don't understand what is meant by 'draw values of the set of parameters' . Again, in line 10 'drawing samples'. Please explain what is meant by this?

Page 8, line 16-17: Here M1 is specified as being the index flood method based on the method of L-moments. But per comment above, I think it would be better to simply specify the model with $\beta\_0=0$ and then use the tools available to compare nested models in a likelihood framework rather than rely on the L-moments methods. Also, given the large range of drainage areas represented in the data set, I think most hydrologists would not consider this case study to qualify as a 'homogeneous region' This is also evident from the athors' figure 3, middle panel, which clearly show that the scale parameter (synonymous with L-CV) to depend on drainage area. As such, M1 is clearly inappropriate for this case. One of the main findings of the study highlighted in the abstract is that the HB-model is superior to the index flood method, but I think this is not a fair comparison.

Section 3.2: I think comparing the results for just two sites is insufficient. I would have expected at least a comprehensive cross-validation (leave-on-out) analysis at this stage.

Figure 5: Interesting to see confidence intervals that drops below zero for the at-site
model. This is clearly inappropriate as also pointed out by the authors. Maybe use this to emphasis the value of the regional model?

Summary: There is a lot of emphasis on future work, which might or might not of course be conducted. In any case, I think there was scope for exploring some of these aspects in the current manuscript at the expense of removing the comparison with the misguided M1 model.

---

## Referee Comment (RC2) · Anonymous Referee #2 · 23 Mar 2018

The paper present a hierarchical Bayesian model for flood frequency analysis and compares its performance against some standard methods for a case study in China. The use of Bayesian Hierarchical models for flood frequency analysis has been proposed elsewhere by several authors and in that sense the paper is not very innovative, although it presents an interesting application. The paper is well structured, although it contains some typos and grammar mistakes, so I would recommend to proof-read the manuscript again. I am not entirely convinced by the evidence the authors bring to support their hypothesis that the model proposed in the model would outperform the "good old" RFA approach in practice: they only show the estimated frequency curves of two stations, without quantifying the gain when Bayesian methods are used.

I have a few major comments regarding the manuscript given below:

1. The different methods compared in the paper partially correspond to different sets of priors: Coles and Dixon (1999) discuss how the L-moment estimation enforces some constrains on the value of the shape parameter, while it is known that Bayesian techniques are somewhat equivalent to MLE methods when diffuse priors are used. The fact that 3 different estimation methods are used for the different pooling approaches makes it really hard to understand whether the (claimed) better performance is due to the estimation is due to the estimation procedure (which is tightly linked with the prior used for the shape parameter) or the pooling strategy. One could do at-site Bayesian estimation with similar priors for the each of the site and devise a Bayesian fully-pooled model to make a comparison which is actually based on the pooling strategy. I appreciate the authors are trying to compare the proposed estimation procedure to the most commonly used approaches, but the comparisons are somewhat spurious. Finally the authors claim that the proposed model performs better than any other method, but I find the evidence to support this statement weak - since it is based mostly on the visual inspection of the flood frequency curves of two stations. More formal criterion could be employed - see for example Kobierska et al. (2017).

2. Page 6 - line 16-17: the model the authors suggest is perfectly valid, although looking at Figure 3 it is obvious that both $\log(\mu)$ and $\log(\sigma)$ change with a similar rate as a function of the log(Area). I wander if the authors could have employed a simpler model (which somehow corresponds to the model used in the index flood approach) by taking $\log(\sigma_i) = k \log(\mu_i)$, with $k \sim N(\bar{\kappa}, \tau_\kappa)$ corresponding to a overall "coefficient of variation" term. Further I appreciate that Area does not seem to have an effect on the value of the shape parameter, but the fact that some of the sample values are positive and some are negative begs the question of whether any other catchment property (steepness? permeability?) could be used to classify catchments with upper or lower bound (and by the way we do not know which parametrisation of the GEV is used in the manuscript). By shrinking all the shape parameters towards some overall average

value, we are forcing things towards a Gumbel distribution, which might be the wrong distribution. This goes back to the question of whether the stations in the analysis can be deemed to be a "homogeneous region" , which essentially corresponds to assuming that the shape parameter can be taken to be the same across all stations.

3. Page 7 - line 6: considering the recommendation in Gelman et al (2017) to prefer weakly informative priors to non-informative ones it is strange to see the Gelman et al (2014) citation to justify the use of diffuse priors. In general I would believe that some knowledge is available about proprieties of the GEV parameters - see for example the "geophysical" prior of Martins and Stedinger (2000). The use of vague non-informative priors can be valid, but much of the recent research in applications seem to indicate that some weakly informative priors might be a better reflection of our current understanding of the properties of parameter values.

4. Section 3.2: somewhere in the results it would be good to have some more information on the MCMC chains and reassurance on the convergence of the MCMC procedure (this is mentioned in page 8 line 5-9, if 15000 sample were taken this seems quite a small number for such a complex problem. What is the equivalent sample size?)

5. What do the posterior for $\alpha_1$ and $\beta_1$ look like? Are they centred around 1? Could these be used to derive some scaled version of the flows? In general I would thing that the posterior of the original parameters would be of interest int he paper (although maybe to be placed in a supplementary material section).

6. Page 10 - line 5. Intervals derived from MLE are typically called confidence not credible intervals (credible intervals are obtained in the Bayesian framework). There is little information given on how the confidence intervals for the MLE estimation are built, my guess is the Delta Method on the quantile itself. One option which might maybe avoid the unpleasant behaviour of the interval covering negative values would be to use the profile likelihood approach or to use the delta method for the log(quantile), since the normal approximation will hardly hold at such high return periods as those shown

in the Figures.

7. Page 11 - line 16-18: there seem to be an indication that the RFA assumes homogeneity, while the HB model does not make this assumption. I would argue that assuming that the shape parameter comes from a common normal distribution is a form of homogeneity assumption.

8. Page 11 - line 19: I do not see why using a HB gives a better way to use short records. In the typical RFA setting shorter records get a smaller weight in the estimation of the regional parameter. I can agree that the Bayesian representation is more elegant, but essentially for short records the final estimated value will simply depend more on the priors and on the other stations in the region. Conversely, stations with short records will be given less weight in the estimation, both in HB and RFA.

9. Table 1: are the the negLik, AIC and BIC of just one series? These quantities can not really be calculated by putting together data across different stations... Further, the AIC/BIC is more often used to compare models with different complexity rather than different distributions, for which tests like the Anderson Darling and Kolmogorov-Smirnov Test are more frequently used.

Minor technical points:

is Equation (4) a likelihood or a posterior? The notation of the formula is not very clear (why the subscript H? is i a parameter?)

Page 7 - line 10: what is the $\beta$ in $\Lambda$? I also guess that if the $\alpha_*$ and $\beta_*$ are included in the vector, one should not have $\mu$ and $\sigma$

Figure 5 and 6: is the y-axis on the log scale? This is not very common - might be worth making it explicit in the axis.

References

Coles, and Dixon (1999). Likelihood-based inference for extreme value models. Extremes.

Gelman, A.; Simpson, D.; Betancourt, M. (2017) The Prior Can Often Only Be Understood in the Context of the Likelihood. Entropy, 19, 555.

Kobierska, Engeland, Thorarinsdottir (2017), Evaluation of design flood estimates - a case study for Norway. Hydrology Research; DOI: 10.2166/nh.2017.068

Martins, E. S., and J. R. Stedinger (2000), Generalized maximum-likelihood generalized extreme-value quantile estimators for hydrologic data, Water Resour. Res., 36(3), 737-744, doi:10.1029/1999WR900330.